# Humoral Immunogenicity of mRNA Booster Vaccination after Heterologous CoronaVac-ChAdOx1 nCoV-19 or Homologous ChAdOx1 nCoV-19 Vaccination in Patients with Autoimmune Rheumatic Diseases: A Preliminary Report

**DOI:** 10.3390/vaccines11030537

**Published:** 2023-02-24

**Authors:** Porntip Intapiboon, Nawamin Pinpathomrat, Siriporn Juthong, Parichat Uea-Areewongsa, Jomkwan Ongarj, Boonjing Siripaitoon

**Affiliations:** 1Department of Internal Medicine, Faculty of Medicine, Prince of Songkla University, Songkhla 90110, Thailand; 2Department of Biomedical Sciences and Biomedical Engineering, Faculty of Medicine, Prince of Songkla University, Songkhla 90110, Thailand

**Keywords:** humoral, immunogenicity, mRNA vaccine, autoimmune rheumatic diseases

## Abstract

Immunogenicity data on the mRNA SARS-CoV-2 vaccine booster after completing a primary series vaccination, other than the mRNA vaccine, in patients with autoimmune rheumatic diseases (ARDs) is scarce. In this study, we reported the humoral immunogenicity of an mRNA booster 90–180 days after completing heterologous CoronaVac/ChAdOx1 nCoV-19 (*n* = 19) or homologous ChAdOx1 nCoV-19 (*n* = 14) vaccination by measuring the anti-SARS-CoV-2 receptor binding domain (RBD) IgG levels at one and three months after mRNA booster vaccination. This study included 33 patients with ARDs [78.8% women; mean (SD) age: 42.9 (10.6) years]. Most patients received prednisolone (75.8%, mean [IQR] daily dose: 7.5 [5, 7.5] mg) and azathioprine (45.5%). The seropositivity rates were 100% and 92.9% in CoronaVac/ChAdOx1 and ChAdOx1/ChAdOx1, respectively. The median (IQR) anti-RBD IgG level was lower in the ChAdOx1/ChAdOx1 group than in the CoronaVac/ChAdOx1 group (1867.8 [591.6, 2548.6] vs. 3735.8 [2347.9, 5014.0] BAU/mL, *p* = 0.061). A similar trend was significant in the third month [597.8 (735.5) vs. 1609.9 (828.4) BAU/mL, *p* = 0.003]. Minor disease flare-ups occurred in 18.2% of the patients. Our findings demonstrated satisfactory humoral immunogenicity of mRNA vaccine boosters after a primary series, with vaccine strategies other than the mRNA platform. Notably, the vaccine-induced immunity was lower in the ChAdOx1/ChAdOx1 primary series.

## 1. Introduction

*Vaccination against COVID-19* is an effective strategy to reduce the severity and mortality of infections with severe acute respiratory syndrome coronavirus 2 (*SARS-CoV-2*) [1]. However, this strategy is limited by the decline in vaccine immunity within a few months after vaccination in healthy and immunocompromised individuals, including those with autoimmune rheumatic diseases (ARDs) [2,3]. Vaccine waning and reduced immunogenicity have been extensively reported in patients with ARDs, compared to healthy people [2,4]. This observed, attenuated humoral immunity in immunosuppressed patients after the primary series vaccination is related to the use of immunosuppressive drugs, especially glucocorticoids, methotrexate, mycophenolate mofetil, rituximab, and abatacept [5,6,7,8]. Nevertheless, some studies have revealed a preserved cellular response after homologous ChAdOx1 nCoV-19 or heterologous inactivated (CoronaVac)/adenoviral vector (ChAdOx1 nCoV-19) vaccination, suggesting the beneficial effect of the adenoviral vector-based vaccine. Therefore, mRNA vaccine booster shots are strongly recommended in various guidelines because two doses of the primary series of SARS-CoV-2 vaccines are insufficient to curb infection spread and prevent morbidity, especially in patients with ARDs [9,10].

Previous studies on COVID-19 vaccine boosters in patients with ARDs have focused on mRNA primary series, and the seropositivity rate varied between 70–93% [11,12,13]; however, only half of these patients gained the optimal neutralizing capacity to the Omicron variants [14,15]. Despite these previous studies, immunogenicity data on primary series other than mRNA vaccines in patients with ARDs are limited. Therefore, the present study aimed to evaluate the humoral immunogenicity of mRNA boosters after heterologous inactivated/adenoviral vector (CoronaVac/ChAdOx1) or homologous adenoviral vector (ChAdOx1/ChAdOx1) vaccination in patients with ARDs. We also evaluated adverse events following immunization (AEFI) and disease activity after immunization.

## 2. Material and Methods

### 2.1. Study Design and Population

This prospective cohort study was conducted at Songklanagarind Hospital, Thailand, from February to April 2022. Patients diagnosed with ARDs who received mRNA vaccine boosters were enrolled. The inclusion criteria were the following: (i) age 18–60 years; (ii) stable disease activity defined by a disease activity score of 28 (DAS28) <5.1 points for rheumatoid arthritis (RA), clinical Systemic Lupus Erythematosus Disease Activity Index (clinical *SLEDAI*) <4 points for systemic lupus erythematosus (SLE), or by a physician judgment for other ARDs; and (iii) received a stable dose of at least one immunosuppressive drug as follows: glucocorticoids equivalent prednisolone < 20 mg/day, methotrexate ≥ 10 mg/week, azathioprine ≥ 50 mg/day, mycophenolate mofetil ≥ 1000 mg/day, or leflunomide (LEF) ≥ 10 mg/day for a month before vaccination. The participants were advised to suspend immunosuppressive drugs according to the American College of Rheumatology (ACR) guidance for SARS-CoV-2 vaccination in patients with ARDs [9]. During the study period, participants who were infected with COVID-19 (discovered by history-taking and elevated anti-receptor binding domain [RBD] IgG in the third month, compared to the first month) were excluded. All participants provided written informed consent prior to participating in the study. This study was approved by the Human Research Ethics Committee, Faculty of Medicine, Prince of Songkla University (REC. 65-001-14-1) and adhered to the principles of the Declaration of Helsinki and Good Clinical Practice.

### 2.2. Vaccination Regimens

Patients with ARDs who previously received the heterologous prime-boost vaccination regimen with the inactivated whole-virus vaccine CoronaVac (Sinovac Biotech, Beijing, China), followed by an adenoviral-vectored vaccine, AZD1222 (ChAdOx1 nCoV-19, AstraZeneca, Oxford, UK), administered 3–4 weeks apart (group 1, CoronaVac/ChAdOx1), or the homologous primary series vaccine regimen with two injections of the ChAdOx1-CoV-19 vaccine, administered 8–12 weeks apart (group 2, ChAdOx1/ChAdOx1), were enrolled in the study. Participants received a third mRNA COVID-19 vaccine booster after their primary series at an interval of 90–180 days, the boosters being either BNT162b2 (Pfizer-BioNTech, Pearl River, NY, USA) 30 μg or mRNA-1273 (Moderna, Norwood, MA, USA) 100 μg, according to vaccine availability. These vaccine regimens were recommended by the Thai National Vaccine Committee in 2021–2022.The CONSORT flow diagrams of vaccine regimens are shown in Figure 1.

### 2.3. Vaccine Humoral Immunogenicity

We evaluated anti-RBD IgG levels at 1 and 3 months after the mRNA vaccine boosters using a chemiluminescent assay against a recombinant spike (S) protein (S1/S2) using the ARCHITECT I System (Abbott Laboratories, Chicago, IL, USA) and chemiluminescent microparticle immunoassay (SARS-CoV-2 IgG II Quant; Abbott Laboratories). The results were expressed as the World Health Organization’s standardized binding antibody unit (BAU/mL). Seropositivity was defined as anti-RBD IgG level ≥42 BAU/mL.

### 2.4. AEFI and Disease Activity after Vaccination

AEFI after the mRNA vaccine boosters was retrospectively surveyed using predefined questionnaires. The reactions were categorized by chronology as immediate or delayed and by severity as local or systemic. ARD activity was monitored from the date of the booster until three months after vaccination. Disease flare-up was defined as an increase in the DAS28 score of >1.2 points or of >0.6 points if the current DAS28 scores were ≥3.2 points for RA, an increase in the clinical *SLEDAI* by ≥4 points for SLE, or an increase in corticosteroid and immunosuppressive drug doses, as determined by the attending rheumatologist.

### 2.5. Statistical Analysis

R version 3.5.1 statistical software (R Foundation for Statistical Computing, Vienna, Austria) was used for the data analysis. The categorical variables were performed using Chi-squared and Fisher’s exact tests and presented as counts and percentages. Comparisons of continuous variables between groups were performed using the *t*-test or Wilcoxon rank-sum tests and presented as the mean (standard deviation [SD]). If data were not normally distributed, they were reported by the median and the interquartile range [IQR] instead. Statistical significance was set at *p* < 0.05.

## 3. Results

### 3.1. Baseline Characteristics

A total of 33 patients with ARDs who received mRNA boosters were enrolled, of whom 19 received the CoronaVac/ChAdOx1 vaccine, and the remaining 14 received the ChAdOx1/ChAdOx1 vaccine. Most of the participants were female (78.8%), and the mean (SD) age was 42.9 (10.6) years. Among the ARD patients, half of the patients were diagnosed with SLE (*n* = 18, 54.5%), followed by RA and spondyloarthritis (*n* = 5, 15.2%), and idiopathic inflammatory myositis (*n* = 4, 12.1%). Glucocorticoids were prescribed in three quarters (*n* = 25, 75.8%), and the median (IQR) prednisolone equivalent dose was 7.5 (5, 7.5) mg/day. Among the immunosuppressive drugs used, azathioprine was the most common (*n* = 15, 45.5%), followed by methotrexate (*n* = 13, 39.4%), mycophenolate mofetil (*n* = 7, 21.2%), and leflunomide (*n* = 3, 9.1%). The baseline characteristics were similar between the CoronaVac/ChAdOx1 vaccine and the ChAdOx1/ChAdOx1 groups (Table 1).

### 3.2. Humoral Immunogenicity of mRNA Vaccine Booster after CoronaVac/ChAdOx1 or ChAdOx1/ChAdOx1 Vaccination

Humoral immunogenicity was measured using anti-RBD IgG levels at 1 and 3 months following mRNA booster vaccination. The values obtained for both groups are shown in Figure 2 and Table 2. The mean (SD) duration between the completed second dose of the primary series and the booster dose was 118 (19.5) days, which was similar between the groups (*p* = 0.886). One month after mRNA booster vaccination, most participants were seropositive (*n* = 35, 96.9%), and only one participant in the ChAdOx1/ChAdOx1 group was seronegative; this patient had SLE and had been taking mycophenolate mofetil. The median (IQR) anti-RBD IgG levels were lower in the ChAdOx1/ChAdOx1 group than in the CoronaVac/ChAdOx1 group (1867.8 [591.6, 2548.6] vs. 3735.8 [2347.9, 5014.0] BAU/mL, *p* = 0.061). After three months, five patients were excluded due to SARS-CoV-2 infection; however, these patients had only mild symptoms. The immunogenicity data at three months post mRNA boosters revealed significantly lower mean (SD) anti-RBD IgG levels in the primary series ChAdOx1/ChAdOx1 group than in the CoronaVac/ChAdOx1 group (597.8 [735.5] vs. 1609.9 [828.4] BAU/mL, *p* = 0.003).

### 3.3. Humoral Immunogenicity of the mRNA Vaccine Booster after CoronaVac/ChAdOx1 or ChAdOx1/ChAdOx1 Vaccination in Immunosuppressive Drug Subgroups

Anti-RBD IgG levels in the immunosuppressive subgroups are shown in Table 3. There were no differences in the 1-month median (IQR) anti-RBD IgG levels between patients with ARDs who received glucocorticoids and patients who did not receive glucocorticoids (2648.7 [774, 4864] vs. 3417.7 [2066.3, 5204.9] BAU/mL, *p* = 0.389). The mean (SD) anti-RBD IgG levels did not differ significantly between the mycophenolate mofetil (2895.8 [2449.7] vs. 3590.2 [2847.2] BAU/mL, *p* = 0.561), methotrexate (4355.5 [2835.6] vs. 2849.8 [2585.1] BAU/mL, *p* = 0.126), and azathioprine groups (3128.1 [2776.1] vs. 3705.3 [2772.9] BAU/mL, *p* = 0.556). Of note, anti-RBD IgG levels tended to be the lowest in patients with ARDs who received mycophenolate mofetil. The same trend was observed in the third month of humoral immunogenicity. The lack of healthy controls and the frequent use of glucocorticoids combined with other immunosuppressive drugs limited our ability to compare the effects of immunosuppressive medications. Nevertheless, in a subgroup analysis of immunosuppressive drugs, 1-month mean (SD) anti-RBD IgG levels in mycophenolate mofetil groups were lower than those in patients with ARDs receiving methotrexate (2895.8 [2449.7] vs. 4355.5 [2835.6] BAU/mL, *p* = 0.266) and azathioprine (2963.8 [2790.5] vs. 3128.1 [2776.1] BAU/mL, *p* = 0.899). However, the differences were not significant, due to the small sample sizes.

### 3.4. Humoral Immunogenicity of MRNA Vaccine Boosters after CoronaVac/ChAdOx1 or ChAdOx1/ChAdOx1 Vaccine Wanes 3 Months after Vaccination

The humoral immunogenicity in the third month was compared to the anti-RBD IgG levels in the first month, and the reduction ratio was calculated. The median (IQR) anti-RBD IgG levels decreased threefold (1.9, 3.7). Moreover, the reduction rate was significantly lower in the homologous ChAdOx1/ChAdOx1 group compared to the reduction rate in the heterologous CoronaVac/ChAdOx1 primary series [3.6 (2.9, 4.7) vs. 2.5 (1.8, 3.3), *p* = 0.032].

### 3.5. Post-Booster Reaction and Disease Activity

The immediate and delayed reactions after the mRNA vaccine boosters were evaluated using the questionnaires shown in Table 4. AEFI occurred in 24 participants (72.7%). Systemic reactions were reported in 24.2% of patients, whereas local reactions occurred in 69.6% of the patients (77.7% in the CoronaVac/ChAdOx1 group and 64.2% in the ChAdOx1/ChAdOx1 group). The most common systemic response was fever (21.2%), whereas pain at the injection site commonly presented as a local reaction (66.7%). The ARD activity was evaluated throughout the study period; disease flare-ups occurred in six patients (18.2%) with increased DAS28 scores and in two patients with RA and psoriatic arthritis, proteinuric flare-ups occurred in a patient with SLE, and increased muscle enzyme levels occurred in a patient with idiopathic inflammatory myositis. However, all six patients had relatively minor disease flares.

## 4. Discussion

During the vaccine shortage in the early spread of SARS-CoV-2, primary series regimens other than mRNA vaccines were implemented in some countries to mitigate this problem. However, many questions arose concerning vaccine immunogenicity after the booster dose and the optimal time for the fourth dose. In the present study, we revealed a satisfactory humoral immunogenicity of the mRNA vaccine booster; however, humoral immunogenicity was lower in homologous ChAdOx1/ChAdOx1 compared to the heterologous CoronaVac/ChAdOx1 primary series vaccination regimen. Immunogenicity declined threefold 3 months after booster vaccination, especially in the homologous ChAdOx1/ChAdOx1 primary series. Although there was no difference in the AEFI between the two groups, the vaccine caused disease flares, posing a concern that needs to be validated in a larger population.

A decline in vaccine immunogenicity has been observed in patients with ARDs receiving immunosuppressive drugs [5,6,7,8]. However, some studies have reported a preserved cellular response after homologous ChAdOx1/ChAdOx1 or heterologous CoronaVac/ChAdOx1 vaccines, suggesting the beneficial effect of the vaccine [6,16]. Moreover, because of the waning of immunogenicity over time and the emergence of the SARS-CoV-2 variants of concern (VOCs), third and fourth vaccinations with mRNA vaccines are now recommended in both healthy and vulnerable populations.

Previous studies on vaccine immunogenicity after mRNA booster vaccinations in patients with ARDs who received a homologous mRNA vaccine primaries series reported strong humoral immunity, especially for the appropriate seroconversion rate [17,18,19]. However, the data should be interpreted with caution, owing to the small sample size and non-defined cut-off protective antibody level. In contrast, some studies reported suboptimal protective immunity after mRNA booster vaccination in patients with ARDs. For example, Gragnani et al. demonstrated a significantly lower anti-RBD IgG level in patients with ARDs, compared to that in healthy individuals, and up to one-third of patients with ARDs who received mycophenolate mofetil or rituximab had a suboptimal humoral response after three homologous doses of the BNT162b2 vaccine [11]. Ferri C et al. reported a similar finding, with 7.8% of patients with ARDs having suboptimal responses (≤70 BAU/mL) [12]. These findings suggest that vaccine immunogenicity data should be cautiously interpreted based on the different diseases, immunosuppressive agents, and the timing of immunity evaluation. Nonetheless, in the present study, under similar conditions and timing of the antibody assessment, we demonstrated at least equal or superior anti-RBD IgG levels, compared to the homologous three-dose mRNA vaccination, especially in the primary heterologous CoronaVac/ChAdOx1 subgroups.

Mycophenolate mofetil is a non-biological immunosuppressive drug that strongly impairs humoral immunogenicity after vaccination [20]. Previous studies have revealed that it impacts the effects of SARS-CoV-2 vaccination, including primary series and mRNA-based vaccine booster schemes. The seroconversion rate and anti-SARS-CoV-2 IgG levels decreased in patients with ARDs receiving mycophenolate mofetil in a dose-dependent manner, especially for a daily dose of >1 gm per day [21]. The impairment of humoral immunogenicity after vaccination in mycophenolate mofetil users can be attributed to the inhibition of B cells. It inhibits inosine monophosphate dehydrogenase, thereby reducing guanine ribonucleotide enzyme levels and hampering dendritic cell activation. We detected a greater decrease in anti-RBD IgG levels in patients with ARDs receiving mycophenolate mofetil than in those receiving other immunosuppressive drugs; however, the differences were not significant, owing to the small sample size, dosage of mycophenolate (1–2 g per day), and temporary pause in drug administration, as per the ACR recommendation for SARS-CoV-2 vaccination in patients with ARDs [9]. Methotrexate is an antirheumatic drug commonly used in inflammatory arthritis. There is evidence that it impairs humoral and cellular immune responses after vaccination [8,22]. In our study, anti-RBD IgG levels in methotrexate users improved; this can be explained by the peri-vaccination drug interruption, similar to the results of randomized controlled trials of patients with rheumatoid or psoriatic arthritis in India, in which anti-RBD antibody titers improved after pausing methotrexate [23]. The temporary interruption of the immunosuppressive drug is a helpful strategy to raise the immunogenicity after vaccination. However, mycophenolate mofetil and methotrexate are anchor drugs that control autoimmune diseases; interrupting these medications might activate disease flare-ups. Theoretically, the mRNA vaccine can stimulate the innate immune response by acting as both an immunogen and an adjuvant, activating endosomal and cytosolic pattern-recognition receptors, such as toll-like receptor (TLR)-3 and TLR-7, and causing type I interferon and inflammatory cytokine production [24]. These responses are similar to the pathogenesis of autoimmune rheumatic diseases, such as SLE and RA. Disease flare-ups might occur in susceptible patients, especially patients with ARDs with a minor or unrecognized active disease and a suspended use of immunosuppressive drugs. A review of prior observational studies of patients with ARDs revealed disease flare-up rates of 0.4% to 20%, with mild to severe disease activity [25]. This contributes to vaccine hesitancy from the perspectives of both patients and physicians. In our study, we obtained a disease flare-up rate of 18.2%. The study included participants who had quiescent disease activity and received low to moderate doses of immunosuppressive drugs. All patients were advised to withhold immunosuppressive drugs (except glucocorticoids), following ACR guidelines [9], and this may explain the high flare-up rate. We believe that optimizing individualized strategies might minimize the risk of disease flare-ups while maximizing the vaccine immune response. Vaccination is administered in a quiescent state, and the optimal duration of the peri-vaccination immunosuppressive drug suspension depends on the disease status. A randomized control trial study is needed to address this point.

Currently, no cut-off protective antibody titer level after vaccination has been well documented because of the variation in the VOCs in different periods of endemic infection. The 1-month protective anti-RBD IgG levels range between 70–2360 BAU/mL, depending on the ability to reduce neutralization titers and the subtype of virus VOCs [12,15,26]. Feng et al. reported protective level thresholds of 264 and 506 BAU/mL for anti-spike and anti-RBD antibodies against symptomatic infection with the Alpha (B.1.1.7) variant [27]. Assawasaksakul et al. showed that only 46% of patients with ARDs had optimal humoral responses to the third BNT162b2 mRNA booster vaccine using a cut-off value of 2360 BAU/mL (corresponding to 90% efficacy for the wild-type vaccine) [15]. In the present study, the 1-month protective rate after the mRNA booster was 73.7–100% in the heterologous CoronaVac/ChAdOx1 group and 28.5–92.9% in the homologous ChAdOx1/ChAdOx1 primary series. After completing the heterologous CoronaVac/ChAdOx1 regimen, the mRNA booster resulted in satisfactory protective antibody levels in the ARD population, despite the fact that specific antibody levels dramatically increased following the mRNA vaccine booster in both healthy and ARDs patients [13,26]. Some studies revealed that these antibodies provided insufficient protection against the emerging SARS-CoV-2 VOCs. Moghnieh et al. reported only 23.1% of vaccine effectiveness against the Omicron VOC infection after the BNT162b2 booster and a loss of the correlation between the anti-spike IgG titer and vaccine effectiveness [27]. Using anti-RBD IgG levels alone may not be sufficient to assess the vaccine effectiveness in the upcoming SARS-CoV2 VOCs.

The higher anti-RBD levels observed in the present study after heterologous CoronaVac/ChAdOx1, compared to the homologous ChAdOx1/ChAdOx1 primary series, at 1 and 3 months after the mRNA booster might be explained by the “mix and match” strategy, which enhances vaccine immunogenicity by targeting various immune cells. A previous study using a BALB/c mouse model reported an improvement in neutralizing antibody (Nab) and Th1 T-cell responses after primary vaccination with adenovirus vector-based vaccines and booster vaccination with other vaccine platforms [28]. Additionally, a proof-of-concept heterologous vaccine study demonstrated superior anti-RBD IgG, Nab, and T-cell reactivity after ChAdOx1-nCoV-19 vaccination boosted by the BNT162b2 vaccine, compared to homologous regimens in both healthy populations and patients with ARDs [29]. In addition, we previously reported that T-cell function was preserved, including IFN-γ^+^ CD4^+^ T cells, IFN-γ^+^ CD8^+^ T cells, and TNFα^+^ CD4^+^ T cells, in patients with ARDs who received heterologous CoronaVac/ChAdOx1 as a primary series vaccine [16]. Mahasirimongkol et al. also demonstrated the benefit of using an inactivated vaccine, followed by the ChAdOx1-nCoV-19 vaccine, compared to homologous regimens with an inactivated vaccine or to an adenoviral vector vaccine in a healthy population [30]. A similar finding was reported by Suntronwong et al. describing a higher anti-RBD level one month after mRNA booster vaccination in a healthy population who received the heterologous CoronaVac/ChAdOx1, compared to the homologous ChAdOx1/ChAdOx1 groups (2930–3049 BAU/mL vs. 1876–3034 BAU/mL) [26]. These findings suggest that the enhanced humoral immunogenicity after mRNA booster vaccination in heterologous regimens can be explained by the preserved T cells in these vaccine regimens. The other theory that demonstrated this finding was antibody breadth from inactivated prime strategies. The chemical component of the inactivated vaccine based on the Wuhan-Hu-1 strain composed of nearly whole proteins recognizes different viral variant spike proteins [31]. Roltgen et al. reported that antibody breadth is better against the viral variants from vaccination than after SARS-CoV-2 infection [32]. In addition, the original antigenic sin (OAS) from inactivated vaccination priming with the ancestral strain causes immune imprinting when under exposure to the new antigen or vaccination. Cross-reactive B-cell memory is recalled in response to a closely related antigen. Thus, subsequent or booster administration induces an intense antibody response [33,34]. In addition to inactivated vaccines, this phenomenon has also been reported for mRNA-based vaccines [32] and might explain the better humoral immunity after the mRNA booster in heterologous inactivated/ChAdOx1 regimens. On the contrary, the negative impact of OAS is that it causes a reduced immune response to the variant virus, compared to the original antigen, as seen in influenza virus infections [35]. This inhibitory effect might reduce the protection and durability of vaccine immunity against VOCs. The attenuated response to the VOC epitopes was reported in infected patients with the prior Wuhan-Hu-1-like antigen vaccination, compared with unvaccinated infected patients [32]. Recently, during the Omicron VOC outspreading, multiple studies demonstrated that the fourth (second booster) dose raised a milder immune response and rapidly waned, compared to the previous vaccination [36,37]. This phenomenon might be explained by the OAS theory. Potentially, an up-to-date, VOC-specific vaccine will be required, and vaccine immunity studies, including plaque reduction neutralization tests, will be needed to evaluate the impact of OAS on the VOC-specific vaccine immunity.

The waning of vaccine immunity is a significant issue. In this study, anti-RBD IgG levels were reduced by more than half between 1 and 3 months after vaccination. Based on our results, the optimal time for the second booster (fourth) dose is after at least 3 months. This finding was consistent with the current United States Centers for Disease Control and Prevention-recommended COVID-19 vaccine update in December 2022, which suggested that a fourth dose should be administered after at least 8 weeks in moderately to severely immunocompromised individuals. Owing to the variation in immunosuppressive drug intensity, we recommend individualized vaccination strategies in patients with ARDs. The monitoring of humoral vaccine immunity might be beneficial for some populations, especially for patients receiving a high dose of mycophenolate mofetil.

This study had some limitations. First, it was limited by the small sample size of patients with ARDs and the lack of healthy participants for comparison. Thus, we could not directly evaluate the effects of immunosuppressive drugs on vaccine immunogenicity. Second, we could not extensively assess the immunogenicity of Nabs and cellular immunogenicity, especially the neutralization capacity on VOCs. Nevertheless, owing to time-sensitive research, these findings demonstrated enhanced immunogenicity after a booster vaccine in immunocompromised patients.

## 5. Conclusions

Our findings demonstrate satisfactory humoral immunogenicity of mRNA boosters after a primary series with vaccine strategies other than the mRNA platform. The SARS-CoV-2 vaccine-induced immunity tended to be lower in the ChAdOx1/ChAdOx1 primary series than in the heterologous CoronaVac/ChAdOx1 regimen. This study suggests that the fourth dose in this subgroup should be considered without delay. However, this finding requires validation in future studies with larger populations.

## Figures and Tables

**Figure 1 vaccines-11-00537-f001:**
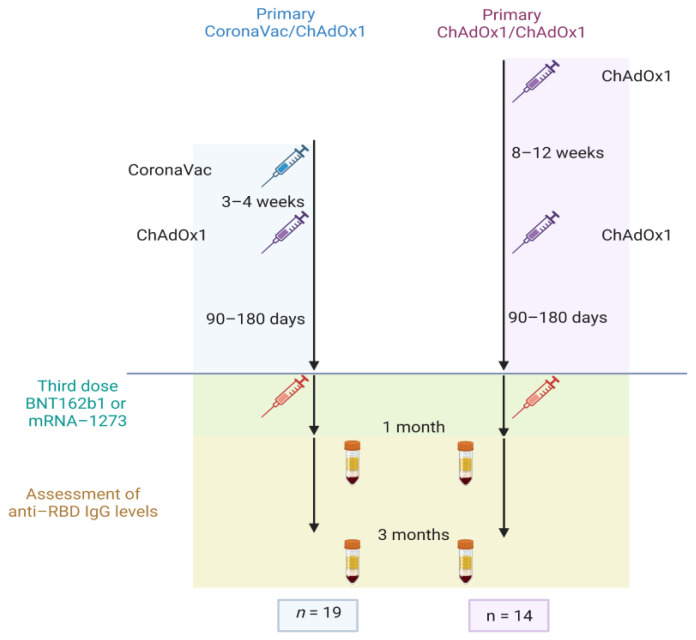
The CONSORT flow diagram of vaccination regimens.

**Figure 2 vaccines-11-00537-f002:**
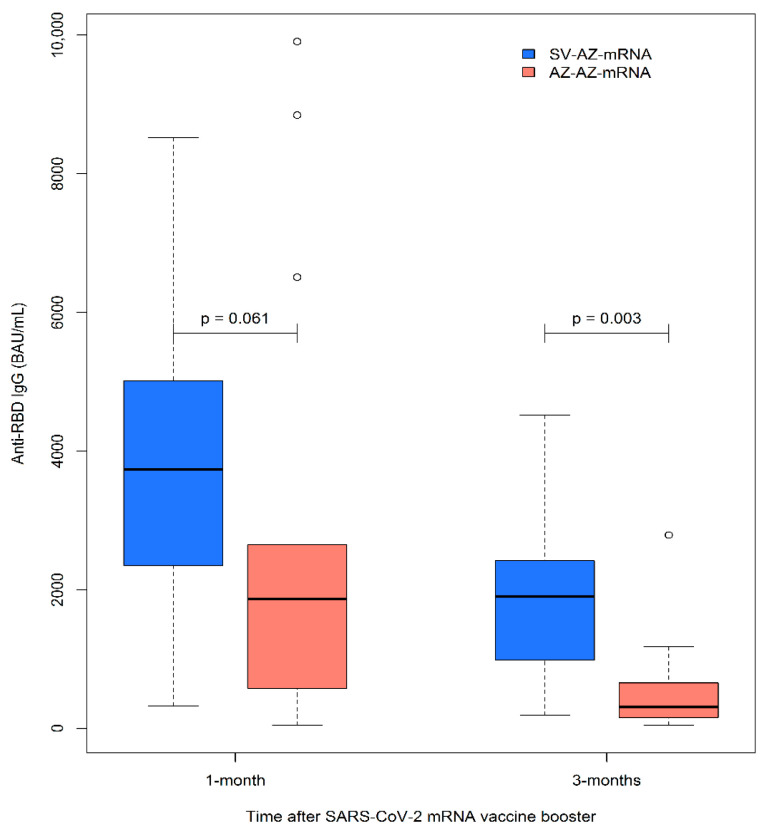
The anti-RBD IgG levels at 1 and 3 months after SARS-CoV-2 mRNA booster vaccination in patients with autoimmune rheumatic diseases who completed heterologous CoronaVac/ChAdOx1 or ChAdOx1/ChAdOx1 vaccination; *p*-value < 0.05 was significant.

**Table 1 vaccines-11-00537-t001:** The baseline characteristics of patients with autoimmune rheumatic diseases who received a SARS-CoV-2 mRNA vaccine booster after completing CoronaVac/ChAdOx1 or ChAdOx1/ChAdOx1 vaccination.

Baseline Characteristics	Total *n* = 33 (%)	CoronaVac/ ChAdOx1 *n* = 19 (%)	ChAdOx1/ChAdOx1 *n* = 14 (%)	*p*-Value
Female sex	26 (78.8)	15 (78.9)	11 (78.6)	1
Mean age, years (SD)	42.9 (10.6)	42.9 (11)	43 (10.4)	0.978
Mean duration (SD), day	117.9 (18.5)	117.5 (18.2)	118.4 (19.5)	0.886
ARDs				0.167
SLE	18 (54.5)	8 (42.1)	10 (71.4)	
RA	5 (15.2)	3 (15.8)	2 (14.3)	
SpA/PsA	5 (15.2)	4 (21.1)	1 (7.1)	
IIM	4 (12.1)	4 (21.1)	0 (0)	
IgG4-Related Disease	1 (3)	0 (0)	1 (7.1)	
GC use	25 (75.8)	13 (68.4)	12 (85.7)	0.416
GC dose, mg (IQR)	7.5 (5, 7.5)	7.5 (5, 7.5)	6.2 (5, 8.1)	0.885
Immunosuppressants				
Azathioprine	15 (45.5)	8 (42.1)	7 (50)	0.923
Methotrexate	13 (39.4)	9 (47.4)	4 (28.6)	0.464
Mycophenolate mofetil	7 (21.2)	4 (21.1)	3 (21.4)	1
Leflunomide	3 (9.1)	3 (21.4)	0 (0)	0.244
Multiple DMARDs	5 (15.2)	5 (26.3)	0 (0)	0.057

ARDs, autoimmune rheumatic diseases; vaccine; DMARDs, disease-modifying antirheumatic drugs; GC, glucocorticoids; IIM, idiopathic inflammatory myositis; PsA, psoriatic arthritis; RA, rheumatoid arthritis; SLE, systemic lupus erythematosus; SpA, spondyloarthritis.

**Table 2 vaccines-11-00537-t002:** The humoral immunogenicity of the SARS-CoV-2 mRNA booster vaccine after completing CoronaVac/ChAdOx1 or ChAdOx1/ChAdOx1 vaccination in patients with autoimmune rheumatic diseases.

Immunogenicity	Total *n* = 33 (%)	CoronaVac/ ChAdOx1 *n* = 19 (%)	ChAdOx1/ ChAdOx1 *n* = 14 (%)	*p*-Value
Mean (SD) time to booster; day	117.9 (18.5)	117.5 (18.2)	118.4 (19.5)	0.886
Seropositivity rate	32 (97.0)	19 (100.0)	13 (92.9)	0.424
1-month median (IQR) anti-RBD IgG (BAU/mL)	2729 (1190.4, 4864)	3735.8 (2347.9, 5014.0)	1867.8 (591.6, 2548.6)	0.061
3-month mean (SD) anti-RBD IgG (BAU/mL) (*n* = 28)	1122.6 (926.5)	1609.9 (828.4)	597.8 (735.5)	0.003 *
Median (IQR) anti-RBD IgG ratio ^1^	3 (1.9, 3.7)	2.5 (1.8, 3.3)	3.6 (2.9, 4.7)	0.032 *

BAU, binding antibody units; ^1^ compared the 3-month anti-RBD IgG level to the 1-month anti-RBD IgG level in the same participant. * *p*-value < 0.05.

**Table 3 vaccines-11-00537-t003:** Summary of 1- and 3-month humoral immunogenicities of SARS-CoV-2 mRNA booster vaccines after CoronaVac/ChAdOx1 nCoV-19 or ChAdOx1/ChAdOx1 vaccination in immunosuppressive drug subgroups.

Immunosuppressive Drugs	1-Month Anti-RBD IgG (BAU/mL)	*p*-Value	3-Month Anti-RBD IgG (BAU/mL)	*p*-Value
Glucocorticoids		0.389		0.297
yes (*n* = 25)	2648.7 (774, 4864)		732.2 (186.8, 1353.6)	
no (*n* = 8)	3417.7 (2066.3, 5204.9)		1558 (387.8, 2015.2)	
Mycophenolate mofetil		0.561		1
yes (*n* = 7)	2895.8 (2449.7)		675.4 (233.4, 2335.3)	
no (*n* = 26)	3590.2 (2847.2)		859.3 (291.6, 1868)	
Methotrexate		0.126		0.162
yes (*n* = 13)	4355.5 (2835.6)		1404 (906.6)	
no (*n* = 20)	2849.8 (2585.1)		897.5 (908.9)	
Azathioprine		0.556		0.228
yes (*n* = 15)	3128.1 (2776.1)		878.7 (771.1)	
no (*n* = 13)	3705.3 (2772.9)		1317.7 (1017.4)	
Comparing immunosuppressiveness				
Mycophenolate mofetil vs. Methotrexate	2895.8 (2449.7) vs. 4355.5 (2835.6)	0.266	1197.5 (1283.6) vs. 1404 (906.6)	0.696
Mycophenolate mofetil vs. Azathioprine	2963.8 (2790.5) vs. 3128.1 (2776.1)	0.899	281.1 (186.4, 1811.3) vs. 732.2 (193.4, 1216.6)	0.815
Methotrexate vs. Azathioprine	4220.4 (2952.9) vs. 2825 (2753)	0.244	1262.3 (928.8) vs. 631.9 (559.1)	0.083

BAU, binding antibody units.

**Table 4 vaccines-11-00537-t004:** Adverse events following immunization and the disease flare rate following SARS-CoV-2 mRNA booster vaccination after completing two doses of CoronaVac/ChAdOx1 or ChAdOx1/ChAdOx1 vaccination.

Adverse Events	Total *n* = 33 (%)	CoronaVac/ChAdOx1 *n* = 19 (%)	ChAdOx1/ChAdOx1 *n* = 14 (%)	*p*-Value
AEFI	24 (72.7)	14 (73.7)	10 (71.4)	1
Systemic reactions	8 (24.2)	4 (21.1)	4 (28.6)	0.695
Fever	7 (21.2)	4 (21.1)	3 (21.4)	1
Chill	3 (9.1)	1 (5.3)	2 (14.3)	0.561
Fatigue	4 (12.1)	1 (5.3)	3 (21.4)	0.288
Myalgia	5 (15.2)	2 (10.5)	3 (21.4)	0.628
Headache	3 (9.1)	1 (5.3)	2 (14.3)	0.561
Local reactions	23 (69.7)	14 (77.7)	9 (64.2)	1
Pain	22 (66.7)	13 (68.4)	9 (64.3)	1
Swelling	3 (9.1)	3 (15.8)	0 (0)	0.244
Disease flares	6 (18.2)	4 (21.1)	2 (14.3)	1
COVID-19 infection	5 (26.3)	5 (26.3)	0 (0)	0.057

AEFI: Adverse events following immunization.

## Data Availability

The study data are available from the corresponding author upon reasonable request.

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
