# Peer review of "Humoral Immunogenicity of mRNA Booster Vaccination after Heterologous CoronaVac-ChAdOx1 nCoV-19 or Homologous ChAdOx1 nCoV-19 Vaccination in Patients with Autoimmune Rheumatic Diseases: A Preliminary Report"

_vaccines, 2023, doi:10.3390/vaccines11030537_

Round 1
Reviewer 1 Report
The aim of this article is to collect and discuss about “immunogenicity data on the mRNA SARS-CoV-2 vaccine booster after completing a primary series vaccination, other than the mRNA vaccine, in patients with autoimmune rheumatic 13 diseases (ARDs)”. The AA evaluated “the humoral immunogenicity of an mRNA booster 90–180 days after completing heterologous CoronaVac/ChAdOx1 nCoV-19 (n=19) or homologous ChA-15 dOx1 nCoV-19 (n=14) vaccination by measuring the anti-SARS-CoV-2 receptor binding domain 16 (RBD) IgG levels at one and three months after mRNA booster vaccination”.
Despite the logical path may be attractive, however, this manuscript is still falling.
1. A comparative analysis with a third group of ADR subjects receiving a third dose of CoronaVac, or ChAdOx1 could give a clearer idea.
2. The protection against the SARS-Cov2 variants represents another limit that has not been taken into account, as below discussed:
“It is known that, despite both naturally infected and vaccinated people showed a high median binding antibody titer against the receptor-binding domain (RBD) of Sp, as well as a production of specific memory T cells however, in both cases, humoral and T-cell responses appear to decline sharply (Pegu A, O’Connell SE et al. Science 2021;373:1372-1377; Falsey AR, Frenck RW Jr, et al. N Engl J Med 2021). A significant waning of vaccine effectiveness against the Variants of Concern (VoCs) within a few months after administration has been shown by retrospective studies performed also on people who have previously received the third dose of a mRNA vaccine (Tal Patalon, et al., Nat. Communic., 2022)”.
Author Response
We thank you for your thoughtful suggestions and insights.
Point 1. A comparative analysis with a third group of ARD subjects receiving a third dose of CoronaVac, or ChAdOx1 could give a clearer idea.
Response 1: Thank you for your comment. However, we could not perform a comparative study with a third dose of CoronaVac or ChAdOx1 nCoV-19 due to the vaccine guideline at that time; the mRNA COVID-19 vaccine was recommended as a booster in immunocompromised hosts, including autoimmune disease patients who received immunosuppressive drugs; I also attached the recommendation for a booster COVID-19 vaccine in autoimmune diseases patients.
1. Curtis, J.R.; Johnson, S.R.; Anthony, D.D.; Arasaratnam, R.J.; Baden, L.R.; Bass, A.R.; Calabrese, C.; Gravallese, E.M.; Harpaz, R.; Kroger, A.; et al. American College of Rheumatology Guidance for COVID-19 Vaccination in Patients With Rheumatic and Musculoskeletal Diseases: Version 4. Arthritis Rheumatol 2022, 74, e21-e36, doi:10.1002/art.42109.
2. Interim statement on the use of additional booster doses of Emergency Use Listed mRNA vaccines against COVID-19 (who.int)
Point 2: The protection against the SARS-Cov2 variants represents another limit that has not been taken into account, as below discussed:
“It is known that, despite both naturally infected and vaccinated people showed a high median binding antibody titer against the receptor-binding domain (RBD) of Sp, as well as a production of specific memory T cells however, in both cases, humoral and T-cell responses appear to decline sharply (Pegu A, O’Connell SE et al. Science 2021;373:1372-1377; Falsey AR, Frenck RW Jr, et al. N Engl J Med 2021). A significant waning of vaccine effectiveness against the Variants of Concern (VoCs) within a few months after administration has been shown by retrospective studies performed also on people who have previously received the third dose of a mRNA vaccine (Tal Patalon, et al., Nat. Communic., 2022)”.
Response 2: Thank you for your suggestion; I agree that the anti-RBD IgG level could not represent the protection against the SARS-Cov2 variants of concern; the neutralization capacity of specific VOCs should be studied to answer this question. However, this is a limitation of this study, and we already add in the limitation discussion. (page 10, line356)
Please see also the attachment ( cover letter, response to comments and revised manuscript)

Reviewer 2 Report
The work present demonstrates the importance of the vaccine strategy that should be used in the study population being evaluated .
Immunogenicity data on SARS-CoV-2 mRNA vaccine booster after completing a primary vaccination series, other than the mRNA vaccine, in patients with rheumatic autoimmune diseases (ARDs) is scarce.
Following the use of mRNA, our results demonstrate satisfactory humoral immunogenicity after a primary series with vaccine strategies other than the mRNA platform. CoV-2 vaccine-induced immunity tended to be lower in the homologous dose series than when administered vaccines in a heterologous scheme . This study suggests that the fourth dose in this population should be considered without delay. However, this finding requires validation in future studies with a larger population with the drugs normally administered
Observation
Abstract: scarse it is not sparse!
In the table 1 in the legends. What means IgG4 RBD, IgG4 related disease ? Add a Ref maybe !
Author Response
We thank you for your thoughtful suggestions and insights.
Point 1 : Abstract: scarse it is not sparse!
Response 1: We are very sorry for our negligence, and we rechecked the text again and change to scares. (page 1, line 14)
Point 2: In the table 1 in the legends. What means IgG4 RBD, IgG4 related disease ? Add a Ref maybe !
Response 2: Special thanks to you for pointing this out. and we have revised the table to IgG4-related disease and deleted it from the legends (page 4, table1)
Please see also the attachment ( cover letter, response to comments and revised manuscript)

Reviewer 3 Report
In this manuscript authors indicated how combination of vaccines and Autoimmune Rheumatic Diseases affect humoral immunity induction. Although this study seems meaningful, there are some points should be addressed as below. Please refer my questions and comments.
1. Why authors use “median (IQR)” for 1-month and “mean (SD)” for 3-month? If you have special reason it should be explained. If not, same way should be used for both.
2. Detailed information for third dose (mRNA vaccine) should be indicated. Or it is possible that choice of mRNA vaccine (BNT162b1 or mRNA-1273) affect humoral immunity and made difference between groups.
3. Minor points
L44: SAR should be SARS
L47: Covid-19 should be COVID-19
L73: Name of the committee should be indicated as full name (including name of institute).
Author Response
We thank you for your thoughtful suggestions and insights.
Point 1 . Why authors use “median (IQR)” for 1-month and “mean (SD)” for 3-month? If you have special reason it should be explained. If not, same way should be used for both.
Response 1: Thank you for your comment. We have already mentioned that “Comparisons of anti-RBD between groups were performed using the T-test or Wilcoxon rank-sum tests, as appropriate ” in statistic analysis. We further explained the statistical analysis as the following; The statistical analysis results from the R program have automatically provided the appropriate statistical test depending on the data variable (normal or not normal distributions) and N of data. The anti-RBD IgG levels in this study were analysed using the student T-test or Ranksum test as appropriate. The 1-month median (IQR) anti-RBD IgG using interquartile range (IQR) due to the spread of the anti-RBD levels had an extreme outlier. In contrast, the 3-month anti-RBD IgG levels were not an extreme outlier, the data were analyzed as mean (SD). (page 3, line 114-117)
Point 2. Detailed information for third dose (mRNA vaccine) should be indicated. Or it is possible that choice of mRNA vaccine (BNT162b1 or mRNA-1273) affect humoral immunity and made difference between groups.
Response 2: Most participants received BNT162b1 mRNA vaccine, and only one received the mRNA-1273 vaccine. Therefore, we could not analyse the impact of the mRNA vaccine subtype on humoral immunity.
Point 3 : Minor points
L44: SAR should be SARS
L47: Covid-19 should be COVID-19
L73: Name of the committee should be indicated as full name (including name of institute).
Response 3: We are very sorry for our negligence. We have already corrected it as per your suggestion.
-L44: SAR should be SARS -> corrected from SAR to SARS (page 2, line 44)
-L47: Covid-19 should be COVID-19 -> corrected from Covid-19 to COVID-19 (page 2, line 47; page 2, line 70)
-L73: Name of the committee should be indicated as full name (including name of the institute). -> We already correct from Human Research Ethics Committee to Human Research Ethics Committee,Faculty of Medicine, Prince of Songkla University (page 2, line 74)
Please see also the attachment (cover letter, response to comments and revised manuscript)

Reviewer 4 Report
This is an interesting study, however, the number of the patients that were included is small n= 19 vs n=14; and thus how can authors be confident that the insignificant p value in Table 1 and Table 3 is due to lack of power or not.
Have you performed power calculation? Please provide more information about that.
In each Table, please provide what statistical test you use and it is mean (SD)?
Author Response
We thank you for your thoughtful suggestions and insights.
Point 1. This is an interesting study, however, the number of the patients that were included is small n= 19 vs n=14; and thus how can authors be confident that the insignificant p value in Table 1 and Table 3 is due to lack of power or not.
Response 1: we agree with your point of view to this extent; this is the major limitation of our study. Due to the long duration of ethical consideration, we could not enrol the cases as the calculated sample size. However, we did our best to analyze the data, and we thought the results of this study were beneficial in a situation with a paucity of immunogenicity studies in autoimmune diseases. A larger study might be needed to answer the effect of immunosuppressive drugs in autoimmune diseases.
Point 2. Have you performed power calculation? Please provide more information about that.
Response 2: In this study, we calculated the sample size by using the study about the durability of antibody response to vaccination general population from McDade et al.’s study. That study revealed a 50.1% reduction of anti-RBD IgG at 3rd-month post-vaccination in 27 healthy participants who received primary series with mRNA vaccine (BNT162b2/Pfizer and mRNA-1273/Moderna) (McDade TW, Demonbreun AR, Sancilio A, Mustanski B, D’Aquila RT, McNally EM. Durability of antibody response to vaccination and surrogate neutralization of emerging variants based on SARS-CoV-2 exposure history. Sci Rep-Uk. 2021;11(1):17325) The output of the sample size calculation from n4Studies: For estimating the finite population proportion Proportion (p) = 0.50, Error (d) = 0.10 Alpha (α) = 0.01, Z (0.995) = 2.575829 Sample size (n) = 24 Allow for 5% missing data: Group1 (m₁) = 25, Group2 (m₂) = 25 Total N =50
Point 3. In each Table, please provide what statistical test you use and it is mean (SD)?
Response 3: We have provided the appropriate statistical test from the R program. The data was analysed using the student T-test, Ranksum test, or Chisq. Test as appropriate. The results were present as mean (SD) or median (interquartile range [IQR]) depending on the data variable (normal or not normal distributions). The statistical test we used to analyze the data is as the following: (page 3, line 114-117)
Table 1
- Baseline characteristic data: Fisher's exact, Chisq.test or student t-test
- GC dose, mg (IQR): Ranksum test
Table 2
- Mean (SD) time to booster: student t-test
- 1-month median (IQR) anti-RBD IgG: Ranksum test
- 3-month mean (SD) anti-RBD IgG: student t-test
Table 3
- Immunosuppressive drugs: Fisher's exact or Chisq. Test
Table 4: Chisq.test or student t-test
Please see also the attachment (cover letter, response to comments, and revised manuscript)

Round 2
Reviewer 1 Report
The aim of this article is to collect and discuss about “immunogenicity data on the mRNA SARS-CoV-2 vaccine booster after completing a primary series vaccination, other than the mRNA vaccine, in patients with autoimmune rheumatic 13 diseases (ARDs)”. The AA evaluated “the humoral immunogenicity of an mRNA booster 90–180 days after completing heterologous CoronaVac/ChAdOx1 nCoV-19 (n=19) or homologous ChA-15 dOx1 nCoV-19 (n=14) vaccination by measuring the anti-SARS-CoV-2 receptor binding domain 16 (RBD) IgG levels at one and three months after mRNA booster vaccination”.
Despite the limits above discussed and partially covered, this manuscript can represent a start point in order to improve the efficacy of vaccination in ARD patients.
Consequently, it is suitable for publication, and we hope that further studies will be carried and that more robust data can be obtained, in order not to frustrate the studies done so far.
Author Response
Thank you for your comments and support